# Research on Supplier Collaboration of Daily Consumer Goods under Uncertainty of Supply and Demand

**Tianwen Song, Qiang Zhang, Junmu Ran and Wenxue Ran \***

Logistics School, Yunnan University of Finance and Economics, Kunming 650221, China; 13924592463@163.com (T.S.); RANJunmu@163.com (J.R.)
**\*** Correspondence: ranwxa@ynufe.edu.cn; Tel.: +86-13608846581

**Abstract:** This paper establishes the decentralized decision-making model of consumer goods, the active collaboration model of consumer goods suppliers, and the decentralized decision-making model of customized consumer goods. Through formula derivation and simulation, the benefit and influence differences of the three modes are compared. This paper studies the influence of supplier's active collaboration on the supply and demand instability of consumer goods and discusses the ideal value of supply chain inventory control. In China's modern circular economy, the inventory control of the consumer goods supply chain is unscientific and nonstandard. We discuss a new inventory control method and study the ideal value of supply chain inventory control. It helps reduce the inventory shortage loss caused by the uncertainty of daily consumer goods, improve the efficiency of the supply chain, enhance the liquidity of supply chain inventory to achieve the purpose of increasing economic value. It has reference value for the inventory control of consumer goods in other countries and regions. The results show that under certain conditions, the horizontal collaboration between suppliers can improve the completeness of consumer goods. The collaboration between suppliers can reduce the hidden dangers caused by the uncertainty of supply and demand and significantly reduce the total inventory cost of manufacturers and improve profits.

**Keywords:** daily consumer goods; uncertain supply; supplier collaboration; horizontal collaboration; inventory cost

## 1. Introduction

With the improvement of people's living standards and the increasing consumption power, the consumer goods industry has also maintained steady or rapid development. At the same time, the large population base and stable growth provide a vast consumer group for the consumer goods industry. Although the consumer goods industry has a vast market space, it is still facing fierce market competition. It has become an essential factor for industry enterprises to maintain core competitive advantage, meet the diversified needs of consumers with low price and high quality, and realize differentiated competition.

Consumer goods have low prices, a high frequency of use, and rich functions, which can meet the increasingly diverse needs of consumers. However, consumer demand is affected by many factors, such as consumer preference, income, commodity price fluctuation, brand effect, etc. Irresistible natural disasters will also lead to the shortage or interruption of the supply of consumer goods. For example, with the spread of the global pandemic, most mask suppliers lack production capacity, and there is a severe shortage of masks in the market. In some areas, masks are out of stock or prices are soaring, resulting in an imbalance between supply and demand. Due to the influence of some uncertain factors, some consumer goods are unsalable or out of stock, which will not only increase the inventory holding cost and out-of-stock cost of production enterprises based on specific common consumer goods, but also lead to the basic needs of consumers cannot be guaranteed. Due to the uncertainty of the supply and demand of consumer goods having a significant impact on the decision-making of the core manufacturers in the supply chain,

and the suppliers of consumer goods being often in the seller's market position, it is difficult to share information with other suppliers of consumer goods easily. Therefore, the active collaboration of daily consumer goods suppliers is essential to reduce logistics costs, increase enterprise profits, and improve customer service level. If some consumer goods are out of stock or delayed due to some uncertain factors, it will increase the inventory holding cost and out-of-stock cost of manufacturers based on specific common consumer goods, and lead to the loss of downstream customers.

Therefore, based on the uncertainty of supply and demand of consumer goods, this paper constructs the decentralized decision-making model of customized consumer goods, the decentralized decision-making model of daily consumer goods, and the active collaboration model of daily consumer goods suppliers. This paper compares the optimal solutions of three models under different conditions, summarizes the application of a dynamic collaboration model of suppliers, improves the overall operational efficiency of the supply chain, improves the stability of consumer goods supply, customer retention rate, customer satisfaction, and the growth rate of enterprises, and ensures the demand of consumers. It can reduce the cost loss and customer loss rate of the manufacturer, improve its operating performance and sales profit. We explore a new supply chain inventory control method to reduce the loss of insufficient inventory caused by the uncertainty and volatility of daily consumer goods, and enhance the liquidity of supply chain inventory, so as to achieve the purpose of increasing economic value.

## 2. Literature Review

The uncertainty of supply and demand is composed of many factors. Ciarallo et al. [1] believe that the production process is uncertain. Chiu et al. [2] proposed that the quality, value, innovation, and unpopularity of brand attributes positively impact customer satisfaction to enhance customer purchase intention. Using brand experience can increase consumers' willingness to pay for price premium [3]. During the period of the COVID-19 pandemic, the demand for PPE surged sharply, and the increase in the use of masks by ordinary people exacerbated the global supply shortage of masks [4–6], which led to the price surge [7,8]. Li et al. [9] analyzed the reasons for the price fluctuation of masks during the epidemic period, including the relationship between short-term mask supply and demand, the imbalance between market supply and demand, the production cost of manufacturers, the hoarding of businesses, and the price fluctuation of masks, which is related to price increase and bid up. Therefore, the increasing complexity of products, the complexity of the manufacturing environment, and the growing emphasis on product quality are the factors leading to the production process's uncertainty.

At the same time, the instability of supply and demand is closely related to supply chain management. At present, supply chain management has become an essential means to enhance enterprises' competitiveness, and the collaborative development of the supply chain has become one of the critical points. Wu et al. [10] objectively studied and analyzed how the collaborative relationship between suppliers affects the supply chain's operation. Zhang et al. [11] proposed a new primary manufacturer supplier collaboration model based on evolutionary game theory and pointed out that manufacturers should fully share information and effectively communicate with suppliers. Compared with a collaborative supply chain, a traditional supply chain can improve the collaborative profit of the supply chain, enhance the core competitiveness of enterprises, and be more robust to the change of lead time [12]. Hofstertter [13] proposed that the creation of organizational structure, the implementation of specific processes and capabilities, the allocation of responsibilities, and the establishment of incentive mechanisms can help companies manage suppliers, in a structured way, product quality and production efficiency. Supplier collaboration at the operational level can improve the enterprise's risk management ability for average risk, internal processing risk, and particular risk to achieve better performance with limited resource input [14]. The supplier collaboration at the operational level can improve the en-

terprise's risk management ability for average risk, internal processing risk, and particular risk to achieve better performance with limited resource investment.

The collaborative supply chain's collaborative relationship is usually divided according to its structure: vertical and horizontal [15]. Traditional research on supply chain collaboration focuses on the vertical collaboration between suppliers, manufacturers, and retailers [16]. More and more attention has been paid to horizontal collaboration in recent years, improving productivity, service level and market position [17], and reducing environmental pollution by reducing transportation distance [18]. Ferrell et al. (2019) [19] made a detailed analysis of horizontal collaboration from the aspects of on-demand logistics, freight integration, facility sharing, incentive mechanism, etc. Zhang et al. (2018) [20] designed a revenue-sharing contract to promote horizontal logistics collaboration in a decentralized environment. With the increase of uncertain demand, horizontal collaboration can reduce working capital investment and improve the filling rate [21]. The current research mainly focuses on the vertical collaboration between manufacturers and suppliers, while the horizontal collaboration between suppliers and suppliers that core manufacturers are responsible for organizing and participating in, needs to be further expanded.

The competition between supply chains is becoming increasingly fierce. How to make their supply chain stand out in such fierce competition is a problem worthy of consideration. The uncertainty of the supply chain leads to the high risk of the supply chain, which often affects the operation of the supply chain. Sreedevi et al. [22] divided supply chain risk into supply risk, manufacturing process risk, and delivery risk. Today's changing business environment is often described as highly competitive, dynamic, and complex [23]. As a part of supply chain management, inventory management also has a significant relationship with uncertain demand. Poor inventory management will be unable to meet the demand, and inappropriate inventory will lead to low efficiency of the supply chain [24]. In supply chain management, managers set the strategies of when to order, order quantity, and the average inventory generated by these inventory replenishment strategies becomes their goal [25]. How to carry out the optimal coordination, replenishment, and inventory control under multiple different supply schemes has always been the concern of researchers and practitioners. Douniel et al. [26] used a limited range Markov decision process to derive the optimal spare parts inventory strategy. Dong et al. [27] developed two modes of mode split transportation (MST) strategy, which integrated inventory control. At the same time, based on consumer demand, inventory control can be realized by calculating the minimum supply chain cost [28]. Based on the characteristics of high flow of consumer goods, FMCGs [29] and Pt. ABC [30] meet customers' needs through the cooperation and integration of the retailer's supply chain. For consumer goods with fluctuating supply-demand relationships [31], the value of the supply chain can be improved by allocating inventory control from the retailer's competitors to the manufacturer [32]. Ahangar et al. [33], taking the municipal solid waste treatment system as the supply chain and the waste as the product, the fuzzy programming, and mixed-integer linear programming model are adopted to achieve the goal of sustainability. Domestic waste and consumer goods have the characteristics of high flow and uncertainty, therefore, these have a good reference value for this paper.

Although inventory is explicitly considered, most of the existing studies do not focus on the possible coordinated inventory control in the supply chain, and ignore the attributes of common use and customization of consumer goods, as well as not considering the horizontal collaborative relationship between suppliers. In view of the above reasons, starting from the uncertainty of the supply and demand of consumer goods, this paper considers the commonality and customization of consumer goods. It constructs a horizontal collaboration model between suppliers of common consumer goods with manufacturers as the core.

## 3. Materials and Methods

According to the above questions, this paper assumes that the supply chain is composed of several suppliers of consumer goods, a single core manufacturer, and customers. Under the condition of uncertain supply and demand, according-to-order (ATO) is adopted. As a mode of production to order in the supply chain, ATO can reduce the complexity of production operation, and has been widely used in the environment of uncertain customer demand. At the same time, considering the universal existence of uncertainty in the supply of consumer goods, based on dividing consumer goods into common consumer goods and customized consumer goods, this paper puts forward a decision-making model of active collaboration among suppliers of common consumer goods, which provides a theoretical basis for horizontal collaboration between suppliers in the uncertain environment of supply and demand.

### 3.1. Model Parameters

Table 1 is a brief description of the symbols used in this paper.

**Table 1.** Definition of Symbols.

| Symbols | Definition | Symbols | Definition |
|---------|-----------|---------|-----------|
| D | Customer demand of finished product | C | Fixed order cost of customized consumer goods |
| P | Price of product | Q | Total orders for customized consumer goods |
| $\gamma$ | Supply factors of customized consumer goods | H | Unit holding inventory cost of customized consumer goods |
| $\kappa$ | Factors of collaborative supply | $C_{supply}$ | Cost of collaborative supply |
| $\mu$ | Unit shortage cost of finished product | $\mu_Y$ | Unit shortage cost of finished product |

This research makes the following assumptions:

(1). According to the different functions of consumer goods in order production, it is assumed that consumer goods can be divided into customized consumer goods and daily consumer goods. In this paper, the daily consumer goods (such as necessities of life) are more general, assuming that they can replace customized consumer goods in the orders' production process. Customized consumer goods are produced according to the needs of customers;

(2). It is assumed that the lead time of consumer goods (common consumer goods and customized consumer goods) is zero;

(3). It is assumed that the customer demand DX of finished product X and the customer demand DY of finished product y obey the orthonormal distribution;

(4). It is assumed that the collaborative supply factors of consumer goods A, B, C, D, and Z obey uniform distribution.

### 3.2. Establishment of Consumer Goods Decision Model under Supply and Demand Uncertainty

Based on the above assumptions and the characteristics of different types of consumer goods, the decentralized decision-making model of customized consumer goods, the decision-making model based on daily consumer goods, and the decision-making model of active collaboration of daily consumer goods suppliers are established.

The decentralized decision-making model of customized consumer goods is the basic model. Based on it, the decentralized decision-making model based on daily consumer goods is obtained by using daily consumer goods instead of customized consumer goods. The consumer goods in both models are in the environment of uncertain supply and demand. This paper optimizes the decentralized decision-making model based on daily consumer goods, considers the collaboration of consumer goods suppliers to solve the problem of supply uncertainty, and then obtains the active collaboration model of daily consumer goods suppliers.

### 3.3. Decentralized Decision-Making Model of Customized Consumer Goods

We constructed an idealized model with only two final products, X and Y, which are produced by two customized consumer goods, A and B, C and D, respectively, as shown in Figure 1.

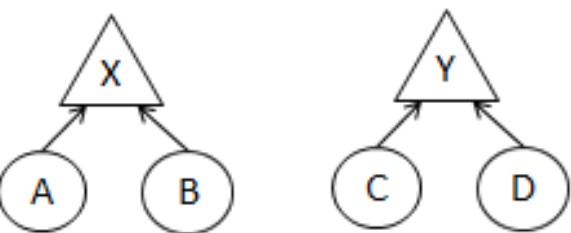

**Figure 1.** Customized consumer goods model.

In the decentralized decision-making model of customized consumer goods, due to the objective existence of supply and demand uncertainty, the actual supply of various customized consumer goods is as follows:

$$S_A = \gamma_A Q_A, S_B = \gamma_B Q_B, S_C = \gamma_C Q_C, S_D = \gamma_D Q_D \tag{1}$$

Therefore, the total inventory cost of the core manufacturers in the supply chain is as follows:

$$TC_1 = \mu_X \max\{[D_X - \min(S_A, S_B)], 0 + \mu_Y \max\{[D_Y - \min(S_C, S_D)], 0\} + H_A \max\{[S_A - \min(D_X, S_A, S_B)], 0\} + H_B$$
$$\max\{[S_B - \min(D_X, S_A, S_B)], 0\} + H_C \max\{[S_C - \min(D_Y, S_C, S_D)], 0\} + H_D \max\{[S_D - \min(D_Y, S_C, S_D)], 0\} + C_A$$
$$+ C_B + C_C + C_D + P_A S_A + P_B S_B + P_C S_C + P_D S_D \tag{2}$$

The first and second items on the right of the equal sign are the out-of-stock costs of finished products X and Y; the third and fourth items are the inventory holding costs of surplus consumer goods A and B due to the mismatch between the actual supply and demand of customized consumer goods A and B; the fifth and sixth items are the surplus consumer goods C due to the mismatch between the actual supply and demand of customized consumer goods C and D. The seventh to tenth items are the fixed order costs of various customized consumer goods; the eleventh to fourteenth items are the purchase costs of various customized consumer goods.

Therefore, the optimal decision-making objective function faced by manufacturers in the supply chain is as follows:

$$\max E(TP_1) = E(P_X \min(D_X, S_A, S_B) + P_Y \min(D_Y, S_C, S_D) - TC_1(Q)) \tag{3}$$

$TP_1$ is the total profit of the consumer goods manufacturer; $E(TP_1)$ is the expected profit; the first and second items in the brackets on the right side of the equation are the sales revenue of finished products X and Y; the third item is the total inventory cost of the manufacturer where $Q = (Q_A, Q_B, Q_C, Q_D)$ is the decision variable of the objective function.

In the traditional decentralized decision-making model of customized consumer goods, manufacturers take their own expected profits as their decision-making objectives. When the parameters meet certain conditions, there will be an optimal total order quantity $Q^* = (Q_A, Q_B, Q_C, Q_D)$, which maximizes the total profit of the objective function.

$TP_1$ in Equation (2) is regarded as the sum of the profits of finished product X and finished product Y. For the order quantity of finished product X, let $Q_1 = Q_A = Q_B$, the expected profit $E(Q_1)$ is expanded in the form of integral, then:

$$E(Q_1) = \int_0^{Q_1} Q_1\{P_X \min(D_X, S_A, S_B) - \mu_X \max\{[D_X - \min(S_A, S_B)], 0\} - H_A \max\{[S_A - \min(D_X, S_A, S_B)], 0\}$$
$$-H_B \max\{[S_B - \min(D_X, S_A, S_B)], 0\} - C_A - C_B - P_A S_A - P_B S_B\}dQ_1 \tag{4}$$

In the above formula, $S_A = \gamma_A Q_A$, $S_B = \gamma_B Q_B$. The first partial derivative of Q1 is obtained by the above formula. When $P_X \to \infty$ or $\mu_X \to 0$ and $Q_1 = Q_A = Q_B \ll D_X$, $Q_1 \to 0$, obviously $\frac{\partial E(Q_1)}{\partial Q_1} > 0$ is tenable; when $H_A$, $H_B \to \infty$ and $Q_1 \gg D_X$, $Q_1 \to \infty$, $\frac{\partial E(Q_1)}{\partial Q_1} < 0$ exists. Therefore, when $P_X$, $\mu_X$, $H_A$, $H_B$, $Q_1$, $D_X$ meet the appropriate relationship, $\frac{\partial E(Q_1)}{\partial Q_1} = 0$. Then the second partial derivative of $Q_1$ is obtained when $P_X \to 0$ or $\mu_X \to \infty$ and $H_A$, $H_B \to 0$, $\frac{\partial^2 E(Q_1)}{\partial Q_1^2} < 0$ exists.

So, the profit of finished product X is a convex function of $Q_1$ when $\frac{\partial E(Q_1)}{\partial Q_1} = 0$ is satisfied, and the expected profit of finished product X reaches the maximum. Similarly, the order situation of finished product Y can be known.

To sum up, when $Q_1$ and $Q_2$ ($Q_2 = Q_C = Q_D$) respectively satisfies the condition that their first partial derivative is 0, then there is an optimal total order quantity $Q^* = (Q_A, Q_B, Q_C, Q_D)$, which maximizes the objective function E ($TP_1$).

*3.4. Decision-Making Model Based on Daily Consumer Goods*

In the decision-making model based on daily consumer goods, daily consumer goods Z is used to replace customized consumer goods B and C, and daily consumer goods Z is shared by finished products X and Y, as shown in Figure 2.

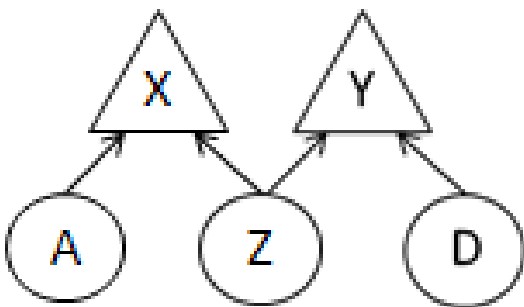

**Figure 2.** Daily consumer goods model.

If the supply of daily consumer goods Z is also uncertain, the distribution principle of consumer goods Z becomes an important factor affecting the manufacturer's expected profit. In this model, the allocation principle of small order first (hereinafter referred to as SOF) is adopted, which means that orders requiring fewer daily consumer goods will be produced first so that manufacturers can deliver their specified finished products to customers as early as possible and improve their response-ability to customers. $Dis_a$ and $Dis_d$ are used to represent the actual quantity of daily consumer goods that can be allocated to orders X and Y so that they can be matched with customized consumer goods A and D. Table 2 shows the dynamic values of $Dis_A$ and $Dis_D$ in different cases.

**Table 2.** Allocation principle of SOF.

| Order size | $Dis_A$ | $Dis_D$ |
|---|---|---|
| $D_X \leq D_Y$ | $Min(D_X, S_A, S_Z)$ | $Min(D_Y, S_D, S_Z - Dis_A)$ |
| $D_X \geq D_Y$ | $Min(D_X, S_A, S_Z - Dis_D)$ | $Min(D_Y, S_D, S_Z)$ |

Based on the daily consumer goods decision model, the total inventory cost of the manufacturer is as follows:

$$TC_2 = \mu_X \max\{[D_X - \min(S_A, Dis_A)], 0\} + \mu_Y \max\{[D_Y - \min(S_D, Dis_D)], 0\} + H_A \max\{[S_A - \min(D_X, S_A, Dis_A)], 0\}$$
$$+ H_Z \max\{[S_Z - Dis_A - Dis_D], 0\} + H_D \max\{[S_D - \min(D_Y, S_D, Dis_D)], 0\} + C_A + C_Z + C_D + P_A S_A + P_Z S_Z + P_D S_D \quad (5)$$

In the above formula, $H_Z$ is the unit inventory holding cost of daily consumer goods Z; $S_Z$ is the actual supply of daily consumer goods Z; $C_Z$ is the fixed order cost of daily

consumer goods Z. Let the total order quantity of daily consumer goods Z be $Q_Z$, and the supply factor be $\gamma_Z$, so $S_Z = \gamma_Z Q_Z$, $Q_Z = Q_A + Q_D$. Based on the daily consumer goods decision-making model, the objective decision-making function faced by the core manufacturers is as follows:

$$\max E(TP_2) = E(P_X \min(D_X, S_A, Dis_A) + P_Y \min(D_Y, S_D, Dis_D) - TC_2(Q)) \qquad (6)$$

In the above formula, $TP_2$ is the total profit of the core manufacturer; the first and second items on the right of the equal sign are the income of finished products X and Y; the third item is the total inventory cost of the manufacturer where $Q = (Q_A, Q_Z, Q_D)$ is the decision variable of the objective function.

In the decentralized decision-making model of daily consumer goods, manufacturers take their own expected profit as the decision-making objective. When the parameters meet certain conditions, there will be an optimal total order quantity $Q^* = (Q_A, Q_Z, Q_D)$, which maximizes the total profit of the objective function.

Therefore, according to Table 2, the distribution principles of daily consumer goods Z are $D_X \leq D_Y$ and $D_X \geq D_Y$. When $D_X \leq D_Y$, we substitute $Dis_A = \min(D_X, S_A, S_Z)$ and $Dis_D = \min(D_Y, S_D, S_Z\text{-}Dis_A)$ into Equation (5). Similarly, $E(TP_2)$ in Equation (5) is regarded as the sum of the profits of finished product X and finished product Y. Then, the total profit of finished product X is a convex function of $Q_A$. When $\frac{\partial E(Q_A)}{\partial Q_A} = 0$ is satisfied, the expected profit of finished product X reaches the maximum. Similarly, the order status of finished product Y can be known. When $D_X \geq D_Y$, there is a similar conclusion.

Therefore, when $Q_A$ and $Q_D$ respectively satisfy the condition that their first partial derivatives are zero, there is an optimal total order quantity $Q^* = (Q_A, Q_Z, Q_D)$, where $Q_Z = Q_A + Q_D$, which maximizes the objective function $E(TP_2)$.

*3.5. Decision Model for Active Collaboration of Daily Consumer Goods Suppliers*

From the above analysis, we can see that the uncertainty of the supply of daily consumer goods will directly affect the decision-making of core manufacturers. Due to the suppliers of daily consumer goods (often key consumer goods) being in a special market position, they will not easily share their private information with other customized consumer goods suppliers. In order to solve this problem effectively and reduce or even avoid the increased inventory cost due to information asymmetry in the supply chain, manufacturers should play their core role in the supply chain. At this time, manufacturers can organize customized consumer goods suppliers to transfer their supply information to daily consumer goods suppliers in real-time and accurately. After daily consumer goods suppliers have enough supply information, they will actively cooperate with customized consumer goods suppliers to supply complete consumer goods. After the active collaboration of daily consumer goods suppliers, the actual supply of daily consumer goods is determined by the sum of the actual supply of two kinds of customized consumer goods, so:

$$S'_Z = \gamma_A Q_A + \gamma_D Q_D \qquad (7)$$

In the active collaboration model of daily consumer goods suppliers, although it can reduce or eliminate the uneven supply of various consumer goods caused by the blocking of supply information, it needs to pay the cost of collaborative supply $C_{supply}$ to support the collaborative operation of the whole supply chain. The effect of collaborative supply is positively correlated with the cost of collaborative supply. Therefore, how to keep a balance between collaborative supply cost and inventory cost saving becomes a compulsory course for core manufacturers:

$$C_{supply} = \kappa \, | \, \gamma_Z Q_Z - \gamma_A Q_A - \gamma_D Q_D \, | \qquad (8)$$

$\kappa$ is the collaborative supply factor. The absolute value represents the difference between the number of finished products actually supplied, and the number of finished

products actually required. In order to minimize this kind of difference, the strategy of active supplier collaboration is adopted.

After the active collaboration of daily consumer goods suppliers, the total inventory cost of core manufacturers is:

$$TC_3 = \mu_X \max[(D_X - S_A), 0] + \mu_Y \max[(D_Y - S_D), 0] + (H_A + H_Z)\max[(S_A - D_X), 0] + (H_D + H_Z)\max[(S_D - D_Y), 0] \\ + C_A + C_Z + C_D + P_A S_A + P_Z S'_Z + P_D S_D + C_{supply} \tag{9}$$

In this case, the optimal decision objective function faced by the manufacturer in the supply chain is as follows:

$$\max E(TP_3) = E(P_X \min(D_X, S_A) + P_Y \min(D_Y, S_D) - TC_3(Q)) \tag{10}$$

In the active collaboration model of daily consumer goods suppliers, manufacturers take their own expected profit as the decisive goal. When the parameters meet certain conditions, there will be an optimal total order quantity $Q^* = (Q_A, Q_Z, Q_D)$ to maximize the manufacturer's expected profit (where $Q_Z = Q_A + Q_D$), and within a certain range of parameters, the collaborative supply model of daily consumer goods suppliers is better than the other two decentralized decision-making models.

From the above analysis, it can be seen that when $Q_A$ and $Q_D$ respectively satisfy the condition that their first partial derivatives are zero, there is an optimal total order quantity $Q^* = (Q_A, Q_Z, Q_D)$, which maximizes the manufacturer's expected profit. By comparing Equations (5) and (9), it can be seen that if the cost of collaborative supply is low and the inventory holding cost of core manufacturers is greatly reduced due to collaborative supply, then the total inventory cost will be reduced. Therefore, the active collaboration model of daily consumer goods suppliers is superior to the decentralized decision model. Considering that the model is too complex to give the exact solution directly, in the simulation and data analysis, through the simulation of the data, the authenticity of the above conclusion is verified, and the optimal numerical solution is given according to the corresponding parameters.

## 4. Results

### 4.1. Parameter Assignment

Due to the complexity, dynamic, and cross nature of the supply chain, this article verifies the rationality of the above three models through mathematical formula derivation and simulation, and compares and analyzes the respective values of the three models through simulation results, so as to provide a reference for suppliers and core manufacturers in the uncertain supply and demand environment. Combined with the actual situation of the consumer goods industry and the relevant data of consumer goods, the settings of the parameters used in this article are shown in Table 3. The data of this paper are from the author's investigation of the supply chain inventory management of a daily consumer goods circulation enterprise in Kunming, Yunnan Province, China.

Under the condition of uncertain supply and demand, the simulation is carried out. By running the simulation function of MathWorks MATLAB r2020b v9.9.0 software, the simulation model is carried out on the Intel Core i7-8550u PC with 16 GB RAM and more than 1.99 ghz CPU.

Finally, the changes of expected profit and expected loss cost are observed and analyzed, and the final conclusion is drawn.

**Table 3.** Parameter value setting table.

| Parameter | Assignment | Parameter | Assignment |
|---|---|---|---|
| $D_X$ | $D_X \sim N(150,15^2)$ | $C_A$ | 1800 |
| $D_Y$ | $D_Y \sim N(150,15^2)$ | $C_B$ | 2400 |
| $P_X$ | 380 | $C_C$ | 1800 |
| $P_Y$ | 450 | $C_D$ | 2400 |
| $Q_A$ | Increase from 200 to 250 in steps of 1 | $C_Z$ | Increase from 1800 to 2400 in steps of 300 |
| $Q_B$ | Increase from 200 to 250 in steps of 1 | $P_A$ | 60 |
| $Q_C$ | Increase from 200 to 250 in steps of 1 | $P_B$ | 90 |
| $Q_D$ | Increase from 200 to 250 in steps of 1 | $P_C$ | 60 |
| $\gamma_A$ | $\gamma_A \sim U(0.7,1)$ | $P_D$ | 90 |
| $\gamma_B$ | $\gamma_B \sim U(0.7,1)$ | $P_Z$ | Increase from 60 to 90 in steps of 15 |
| $\gamma_C$ | $\gamma_C \sim U(0.7,1)$ | $H_A$ | 18 |
| $\gamma_D$ | $\gamma_D \sim U(0.7,1)$ | $H_B$ | 20 |
| $\gamma_Z$ | $\gamma_Z \sim U(0.7,1)$ | $H_C$ | 20 |
| $\mu_X$ | Increase from 200 to 300 in steps of 50 | $H_D$ | 24 |
| $\mu_Y$ | Increase from 220 to 320 in steps of 50 | $H_Z$ | Increase from 18 to 24 in steps of 3 |
| $\kappa$ | Increase from 50 to 100 in steps of 25 | | |

Note: The assignment of $Q_A$ and $Q_D$ in the third model will be explained separately.

### 4.2. Decentralized Decision-Making Model Simulation of Customized Consumer Goods

Suppliers of customized consumer goods A, B, C, and D supply the consumer goods they need based on the manufacturer's total order quantity $Q_A$, $Q_B$, $Q_C$, and $Q_D$ under the condition of asymmetric information in the supply chain.

According to the parameter data in Table 3, the optimal decision variable $Q^* = (Q_A^*, Q_B^*, Q_C^*, Q_D^*)$. The simulation results are shown in Figure 3, and the expected profit is a convex function varying with the order quantity.

According to Table 4, it shows the change of the optimal value of the manufacturer's order quantity when the shortage cost increases from low to high in the customized consumer goods model, which is verified by the first model.

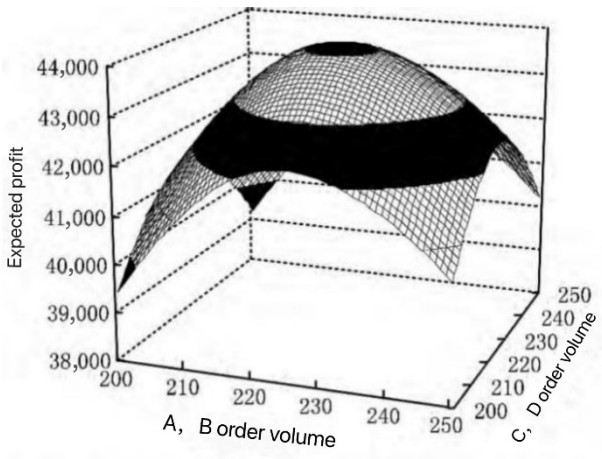

**Figure 3.** Simulation diagram of customized consumer product model ($\mu_X = 200$, $\mu_Y = 220$, record a time of 132 s).

**Table 4.** Optimal values of customized consumer product models.

| $\mu_X$ | $\mu_Y$ | $(Q_A^*, Q_B^*, Q_C^*, Q_D^*)$ | Income | Cost | Profit |
|---|---|---|---|---|---|
| 200 | 200 | (225, 225, 228, 228) | 109,891 | 65,761 | 44,130 |
| 250 | 270 | (227, 227, 230, 230) | 110,247 | 66,637 | 43,610 |
| 300 | 320 | (229, 229, 232, 232) | 110,575 | 67,447 | 43,128 |

### 4.3. Decentralized Decision-Making Model Simulation Based on Daily Consumer Goods

The suppliers of customized consumer goods *A* and *D* supply the consumer goods they need under the condition of asymmetric information in the supply chain according to the manufacturer's total order quantity, $Q_A$ and $Q_D$. According to the parameter data in Table 3, the optimal decision variable $Q^* = (Q_A^*, Q_Z^*, Q_D^*)$. The simulation results are shown in Figure 4. The expected profit is a convex function that varies with the order quantity of A and D.

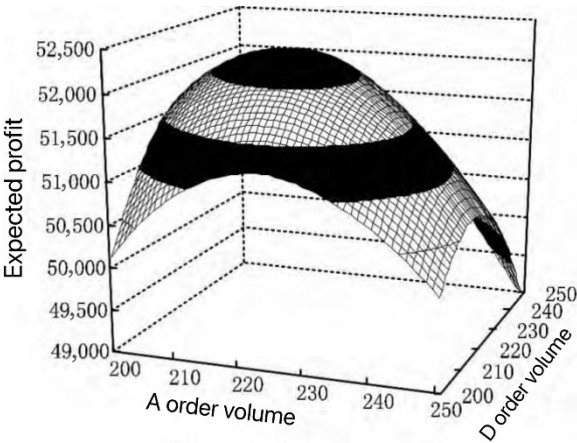

**Figure 4.** Simulation diagram of daily consumer goods model (with low relevant cost, $\mu_X = 200$, $\mu_Y = 220$, record a time of 1095 s).

Table 4 shows the change of the optimal value of the manufacturer's order quantity when the relevant cost and shortage cost increase from low to high. The second model is verified.

### 4.4. Active Collaboration Model Simulation for Active Collaboration of Daily Consumer Goods Suppliers

After using common consumer products, the core manufacturer organizes multiple customized consumer product suppliers to accurately share the supply information of customized consumer products with commonly used consumer product suppliers in real-time so that they can collaborate to provide core manufacturers with a complete set of consumer products. According to the parameters in Table 3, the simulation is carried out, and the optimal decision variable $Q^* = (Q_A^*, Q_Z^*, Q_D^*)$ is obtained. In this model, the manufacturer organizes suppliers for collaboration, therefore, the manufacturer does not have to deliberately increase the order volume of A and D to meet customer needs, so $Q_A^*$ and $Q_D^*$ are set to increase from 180 to 230 in steps of 1. The simulation result is shown in Figure 5. The expected profit is a convex function that varies with the amount of A and D orders. It can be seen from Figure 5 that within a certain range, the supplier collaborative supply model is better than the other two models, so the third model is verified.

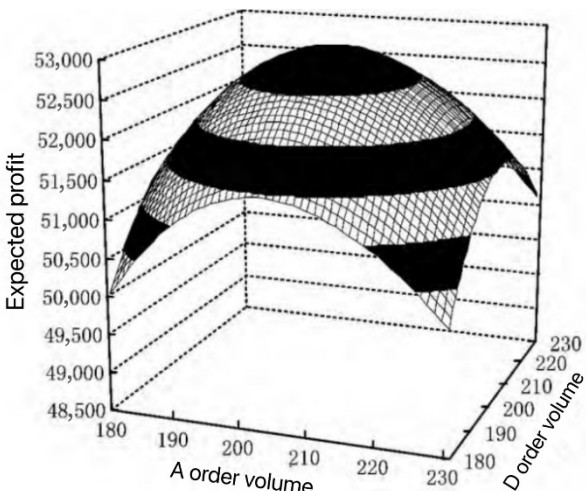

**Figure 5.** Simulation diagram of commonly used consumer product models after collaboration ($\kappa$ = 50, relatively low cost, $\mu_X$ = 200, $\mu_Y$ = 220, record a time of 1124 s).

Table 5 shows the variation of collaborative profit with collaborative supply cost in the active collaborative model of daily consumer goods suppliers.

**Table 5.** Optimal values based on daily consumer goods model.

| Related Costs of Daily Consumer Goods | H$_Z$ | P$_Z$ | C$_Z$ | $\mu_X$ | $\mu_Y$ | $(Q_A^*, Q_Z^*, Q_D^*)$ | Income | Cost | Profit |
|---|---|---|---|---|---|---|---|---|---|
| | | | | 200 | 220 | (226, 444, 218) | 110,559 | 58,286 | 52,273 |
| Lower cost | 18 | 60 | 1800 | 250 | 270 | (229, 449, 220) | 110,959 | 59,107 | 51,852 |
| | | | | 300 | 320 | (231, 453, 222) | 111,262 | 59,789 | 51,473 |
| | | | | 200 | 220 | (222, 436, 214) | 109,850 | 63,379 | 46,471 |
| Moderate cost | 21 | 75 | 2100 | 250 | 270 | (225, 442, 217) | 110,389 | 64,424 | 45,965 |
| | | | | 300 | 320 | (228, 448, 220) | 110,888 | 65,364 | 45,524 |
| | | | | 200 | 220 | (218, 429, 211) | 109,180 | 68,408 | 40,772 |
| Higher cost | 24 | 90 | 2400 | 250 | 270 | (221, 435, 214) | 109,766 | 69,586 | 40,180 |
| | | | | 300 | 320 | (224, 441, 217) | 110,311 | 70,651 | 39,660 |

We can draw the following conclusions by observing the data in Table 6:

(1). For a specific collaborative supply factor, when the inventory holding cost, fixed order cost, and price of daily consumer goods are set, the higher the out-of-stock cost is, the larger the optimal order quantity the manufacturer needs to order, and the corresponding revenue and price will also increase, but the collaborative supply cost will gradually increase, and the profit will decrease;

(2). For a specific collaborative supply factor, the higher the relevant costs (inventory holding cost, fixed order cost, and shortage cost), the less the manufacturer is willing to pay for daily consumer goods. Therefore, manufacturers can only make up for the related costs by reducing the total orders of consumer goods, which will reduce the cost of collaborative supply, but the reduction of orders and the sharp increase of related expenses will reduce the revenue and profit of manufacturers at the same time;

(3). When the related cost and shortage cost of daily consumer goods is fixed, the larger the collaborative supply factor is, the higher the collaborative supply cost the manufacturer has to pay. Simultaneously, the revenue decreases with the decrease of order quantity, which will eventually lead to a decrease in profit.

Table 6. Daily consumer goods model after supplier collaborative supply.

| κ | Related Costs of Daily Consumer Goods | $H_Z$ | $P_Z$ | $C_Z$ | $\mu_X$ | $\mu_Y$ | $(Q_A^*, Q_Z^*, Q_D^*)$ | Income | Cost | Profit | Cooperative Supply Cost |
|---|---|---|---|---|---|---|---|---|---|---|---|
| 50 | Lower cost | 18 | 60 | 1800 | 200 | 220 | (207, 410, 203) | 109,559 | 56,650 | 52,909 | 2432 |
| | | | | | 250 | 270 | (210, 415, 205) | 109,974 | 57,619 | 52,355 | 2461 |
| | | | | | 300 | 320 | (213, 420, 207) | 110,357 | 58,506 | 51,851 | 2491 |
| | Moderate cost | 21 | 75 | 2100 | 200 | 220 | (202, 403, 201) | 108,947 | 61,339 | 47,608 | 2390 |
| | | | | | 250 | 270 | (206, 409, 203) | 109,481 | 62,509 | 46,972 | 2426 |
| | | | | | 300 | 320 | (209, 414, 205) | 109,901 | 63,498 | 46,403 | 2455 |
| | Higher cost | 24 | 90 | 2400 | 200 | 220 | (199, 397, 198) | 108,347 | 65,951 | 42,396 | 2354 |
| | | | | | 250 | 270 | (202, 402, 200) | 108,840 | 67,158 | 41,682 | 2384 |
| | | | | | 300 | 320 | (205, 408, 203) | 109,401 | 68,364 | 41,037 | 2420 |
| 75 | Lower cost | 18 | 60 | 1800 | 200 | 220 | (206, 409, 203) | 109,481 | 57,784 | 51,697 | 3638 |
| | | | | | 250 | 270 | (209, 414, 205) | 109,901 | 58,775 | 51,126 | 3683 |
| | | | | | 300 | 320 | (212, 419, 207) | 110,290 | 59,682 | 50,608 | 3728 |
| | Moderate cost | 21 | 75 | 2100 | 200 | 220 | (201, 401, 200) | 108,752 | 62,337 | 46,415 | 3567 |
| | | | | | 250 | 270 | (205, 408, 203) | 109,401 | 63,639 | 45,762 | 3629 |
| | | | | | 300 | 320 | (208, 412, 204) | 109,733 | 64,556 | 45,177 | 3665 |
| | Higher cost | 24 | 90 | 2400 | 200 | 220 | (198, 395, 197) | 108,136 | 66,914 | 41,222 | 3514 |
| | | | | | 250 | 270 | (201, 401, 200) | 108,752 | 68,260 | 40,492 | 3567 |
| | | | | | 300 | 320 | (204, 406, 202) | 109,219 | 69,388 | 39,831 | 3612 |
| 100 | Lower cost | 18 | 60 | 1800 | 200 | 220 | (204, 406, 202) | 109,219 | 58,731 | 50,488 | 4815 |
| | | | | | 250 | 270 | (208, 512, 204) | 109,733 | 59,833 | 49,900 | 4887 |
| | | | | | 300 | 320 | (211, 417, 206) | 110,135 | 60,765 | 49,370 | 4946 |
| | Moderate cost | 21 | 75 | 2100 | 200 | 220 | (201, 401, 200) | 108,752 | 63,526 | 45,226 | 4756 |
| | | | | | 250 | 270 | (204, 406, 202) | 109,219 | 64,662 | 44,557 | 4815 |
| | | | | | 300 | 320 | (207, 411, 204) | 109,656 | 65,701 | 43,955 | 4875 |
| | Higher cost | 24 | 90 | 2400 | 200 | 220 | (198, 394, 196) | 108,018 | 67,967 | 40,051 | 4673 |
| | | | | | 250 | 270 | (201, 400, 199) | 108,643 | 69,339 | 39,304 | 4744 |
| | | | | | 300 | 320 | (203, 405, 202) | 109,136 | 70,507 | 38,629 | 4803 |

## 5. Discussion

After three different models have been established, we discuss the following.

For customized consumer goods, it can be seen from Table 4 that in order to solve the problem of increasing shortage cost, core manufacturers can only continuously increase the total order volume of consumer goods. However, because of the uncertainty of the supply of consumer goods, although the increase of orders can improve income, it cannot be ignored that the manufacturer's total inventory cost also increased correspondingly, but the final total profit decreased. This shows that we cannot simply increase the order quantity. It will be counterproductive and worsen the impact of uncertainty.

In the model simulation of daily consumer goods, after adopting the decentralized decision-making based on daily consumer goods—regardless of the relevant cost—in order to solve the problem of increasing shortage cost, manufacturers can only increase the revenue of products by continuously increasing the total amount of orders. However, due to the objective existence of uncertainty in the supply of consumer goods, it will increase the total inventory cost of the core manufacturers and then reduce the profits.

Comparing Table 5 with Table 4, it can be concluded that if the inventory holding cost, fixed order cost, and price of daily consumer goods are low or moderate, the strategy based on daily consumer goods can significantly reduce the manufacturer's cost and improve the income and profit; at this time, the strategy based on daily consumer goods is better than the decentralized decision-making of customized consumer goods; if the inventory holding cost, fixed order cost and price of daily consumer goods are low or moderate, the strategy based on daily consumer goods can significantly reduce the manufacturer's cost and improve the income and profit, and when the ordering cost is relatively high, the decision-making based on daily consumer goods will increase the manufacturer's cost and reduce the manufacturer's income and profit due to the higher related cost of daily consumer goods. The decentralized decision-making of customized consumer goods is better than the decision-making based on daily consumer goods.

Under the supplier collaborative supply, comparing Table 6 with Tables 4 and 5, it can be seen that as long as the relevant costs of daily consumer goods (inventory holding cost, fixed order cost, and shortage cost) are high, no matter what the value of collaborative supply factor is, the shortage cost reduced by supplier collaboration cannot offset the higher cost of purchasing daily consumer goods that manufacturers have to pay. At this time, customized consumer goods are decentralized decision-making. The model is better than the supplier active collaboration model. When the collaborative supply factor is small ($\kappa = 50$), and whether the cost of daily consumer goods is high or low, or when the co-supply factor is moderate ($\kappa = 75$), and the price of daily consumer goods is high, the supplier active collaboration model is better than the decision model based on daily consumer goods.

According to the simulation results and data of the above three models, it can be seen that when the related cost of daily consumer goods (inventory holding cost, fixed order cost, and shortage cost) is low or moderate, the daily consumer goods strategy can significantly reduce the inventory cost of manufacturers. However, when the related cost of daily consumer goods is high, decentralized decision-making based on daily consumer goods is often not as effective as customized consumer goods. Therefore, the higher the related cost of daily consumer goods, the lower the value of decentralized decision-making of daily consumer goods, or even lower than that of customized consumer goods. The more significant the importance of active supplier collaboration strategy is, the more significant the impact of collaborative supply factors should also be considered.

Under other specific conditions, to improve the collaborative value and bring more profits to manufacturers, it is necessary to reduce the related costs of collaborative supply factors or daily consumer goods as much as possible so that the enthusiasm and initiative of manufacturers to implement collaborative supply will be stronger and stronger.

## 6. Conclusions

This paper studied the supply chain inventory control in China's current circulation economy, which has a substantial promotion value. The analysis of China's current commodity circulation economy has great practical significance, especially for the inventory control of the supply chain of consumer goods. In this paper, the simulation method is used to verify and study the three models. However, these three models are still in the theoretical stage and have not been applied to inventory control management. In this assumption, the supply factor of consumer goods is set as uniform distribution, and the demand of final customers for finished products is set as a normal distribution. In the uncertain environment of product supply and customer demand, an active cooperation model between daily consumer goods suppliers and multiple customized consumer goods suppliers is constructed under the guidance of core manufacturers. In the high cost of daily consumer goods, decentralized decision-making of customized consumer goods is more critical than decentralized decision-making based on daily consumer goods.

In addition, horizontal cooperation between suppliers can improve the completeness of consumer goods, significantly reduce the total inventory cost of manufacturers, and

improve profits. It is assumed that the collaborative supply cost of horizontal collaboration is less than the difference between the total inventory cost of the manufacturer before and after collaboration. In this case, active supplier collaboration is feasible. The synergy value (profit) is negatively correlated with the synergy supply factor, and the synergy effect is the best when the synergy supply factor is the smallest.

In terms of limitations, this study was conducted in the China consumer goods sector, which has regional limitations. However, this method has an excellent referential value for inventory control research of the world's consumer goods, and it is also suitable for other countries to study in different fields' inventory control issues. In addition, in the hypothesis, the supply factors of consumer goods are set to a uniform distribution, and the customer demand of final products is set to normal distribution. In order to simplify the model data, the lead time of consumer goods is set to zero, and the three models in this paper have not been used in actual production. The above points are the limitations of this paper. Future research can investigate new inventory control methods in other countries and fields, and consider applying them to product inventory control in other high-circulation areas. We can also increase the influence factors of different suppliers' lead times, integrate the interests of the whole supply chain, and let manufacturers punish suppliers who do not supply consumer goods in time. Furthermore, consider letting suppliers share the benefits of cooperation to better participate in the horizontal supply cooperation of consumer goods.

**Author Contributions:** All authors contributed equally to this article. All authors have read and agreed to the published version of the manuscript.

**Funding:** This research was supported by the National Natural Science Foundation of China of Projects No. 71661029.

**Institutional Review Board Statement:** Not applicable.

**Informed Consent Statement:** Not applicable.

**Data Availability Statement:** The data presented in this study are available on request from the corresponding author. The data are not publicly available due to [the enterprise which provides the price data are reluctant to disclose their identities].

**Acknowledgments:** This work was supported by the National Natural Science Foundation of China of Projects No. 71661029.

**Conflicts of Interest:** The authors declare no conflict of interest.

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
