# Peer review of "Research on Supplier Collaboration of Daily Consumer Goods under Uncertainty of Supply and Demand"

_sustainability, doi:10.3390/su13105683_

Round 1
Reviewer 1 Report
Thanks for the opportunity to review this interesting article. However, the article still needs some elements of clarification and improvement as follows:
1. The abstract does not clearly state the purpose or purpose of this article. I suggest the authors to do this!
2. The citation method is not the one agreed by Sustainability magazine! I suggest the authors to adapt to the correct citation mode!
3. The section dedicated to Literature review is very brief and does not specify some aspects meant to clarify the purpose of the research.
4. The section dedicated to research methodology is unclear. After presenting the three models and their simulation, what is the database used for this? What is the source of the data used for the simulation and with what software was it made?
5. The conclusions do not show the economic or managerial implications by simulating the 3 models used by the authors. The authors do not specify in which industrial branch they applied the 3 models or if they considered maybe another economic branch?
Author Response
Thank you for your suggestions, I have revised them in the manuscript. I really appreciate your efforts!
Reviewer 2 Report
Research on Supplier collaboration of daily consumer goods 2 under uncertainty of supply and demand
Based on the characteristics of uncertain supply and demand of consumer goods, this paper focuses on the uncertainty of supply and customer demand. It discusses the influence of active supplier collaboration under unstable supply and demand of daily consumer goods. The decentralized decision-making model of daily consumer goods, the dynamic collaboration model of daily consumer goods suppliers, and the decentralized decision-making model of customized consumer goods are established. The profit and difference of the three models are compared by formula deri vation and simulation
- This is an interesting piece of “Supplier collaboration of daily consumer goods 2 under uncertainty of supply and demand” work. Please underscore the scientific value added/contributions of your paper in your abstract and introduction and address your debate shortly in the abstract.
- Why is the topic important (or why do you study on it)? What are research questions?? What are your contributions? Why is to propose this particular method
- The performance literature review is extensive and covers the relevant material with a considerable degree of insight. The paper is very well structured. The material is well presented I would suggest the author to discuss these references in your context and references. (1) Shahin Sadeghi Ahangar, Amirhossein Sadati & Masoud Rabbani (2021) Sustainable design of a municipal solid waste management system in an integrated closed-loop supply chain network using a fuzzy approach: a case study, Journal of Industrial and Production Engineering, DOI: 1080/21681015.2021.1891146; (2) Hossein Beiki, Seyed Mohammad Seyedhosseini, Vadim V. Ponkratov, Angelina Olegovna Zekiy & Sergei Anatolyevich Ivanov (2021) Addressing a sustainable supplier selection and order allocation problem by an integrated approach: a case of automobile manufacturing, Journal of Industrial and Production Engineering, DOI: 10.1080/21681015.2021.1877202; (3) do Canto, N.R., Bossle, M.B., Vieira, L.M.and De Barcellos, M.D. (2020), "Supply chain collaboration for sustainability: a qualitative investigation of food supply chains in Brazil", Management of Environmental Quality, Vol. ahead-of-print No. ahead-of-print. https://doi.org/10.1108/MEQ-12-2019-0275
- Why do you propose “uncertainty of supply and demand” and “supplier collaboration” as your major concern?
- The references need to update to 2021. This article is referred the SC disruption and organizational ambidexterity “Bui, TD., Tsai, FM., Tseng, ML.*, Tan. R.R., Yu, KDS., Lim, MK. (2021). Sustainable supply chain management towards disruption and organizational ambidexterity: a data driven bibliometric analysis. Sustainable Production and Consumption 26, 373-410”
- I would suggest the discussion and result separate into 2 sections
- Your conclusions' section needs to underscore the scientific value added of your paper, and/or the applicability of your findings/results, as indicated previously. Basically, you should enhance your findings, limitations, underscore the scientific value added of your paper, and/or the applicability of your contributions/shortages and future study in this session.
The literature review should be discussed more. The developed model is quite well described, and appears quite impressive. The findings and results of the case study are also very impressive. The results are clearly analyzed and well argued. The discussion and conclusions are clear and persuasive.
Author Response

(The authors gave the same response as above.)

Reviewer 3 Report
The paper is interesting and deals with a very interesting topic. I have just a few suggestions.
Introduction
Authors should emphasize more the objective of the study and the theoretical contribution offered.
Literature review
This section seems a little weak to me. Authors should extend the literature by adding further studies (the weakness of this section is also demonstrated by the very limited number of references).
Conclusions
Authors should extend the conclusions, emphasize better the contribution to practice and theory. In addition, they should include research limitations and extend insights for future research.
Author Response

(The authors gave the same response as above.)

Round 2
Reviewer 1 Report
Thank you for doing my suggestions!
Reviewer 2 Report
to be accepted